# Bacterial Lysates Modulate Human Macrophage Responses by Inducing BPI Production and Autophagy

**DOI:** 10.3390/biom15101446

**Published:** 2025-10-13

**Authors:** Yael García-González, María Teresa Herrera, Esmeralda Juárez, Teresa Santos-Mendoza, Yolanda González, Eduardo Becerril-Vargas, Silvia Guzmán-Beltrán

**Affiliations:** 1Departamento de Investigación en Microbiología, Instituto Nacional de Enfermedades Respiratorias Ismael Cosío Villegas, Calzada de Tlalpan 4502, Sección XVI, Alcaldía de Tlalpan, Mexico City 14080, Mexicoygonzalezh@iner.gob.mx (Y.G.); 2Laboratorio de Alta Contención Biológica (LACBio), Instituto Nacional de Enfermedades Respiratorias Ismael Cosío Villegas, Calzada de Tlalpan 4502, Sección XVI, Alcaldía de Tlalpan, Mexico City 14080, Mexico; ejuarez@iner.gob.mx; 3Laboratorio de Transcriptómica e Inmunología Molecular, Instituto Nacional de Enfermedades Respiratorias Ismael, Cosío Villegas, Calzada de Tlalpan 4502, Sección XVI, Alcaldía de Tlalpan, Mexico City 14080, Mexico; 4Laboratorio de Microbiología Clínica, Instituto Nacional de Enfermedades Respiratorias Ismael Cosío Villegas, Calzada de Tlalpan 4502, Sección XVI, Alcaldía de Tlalpan, Mexico City 14080, Mexico; edobec.var@gmail.com

**Keywords:** bacterial lysates, bactericidal/permeability-increasing protein, autophagy, *Mycobacterium tuberculosis*

## Abstract

Bacterial lysates have emerged as promising immunomodulatory agents that can enhance innate immune responses. Given the crucial role of macrophages in recognizing and controlling intracellular pathogens such as *Mycobacterium tuberculosis,* this study aimed to evaluate the immunological effects of selected bacterial lysates on human monocyte-derived macrophages (MDMs). We examined the ability of commercial bacterial lysates, Pulmonarom, Ismigen, Uro-Vaxom, and a lysate of *M. tuberculosis* H37 Ra (LMtb) to stimulate the production of key pro-inflammatory cytokines, including TNF-α, IL-1β, and IL-8. In addition, we investigated whether these lysates could modulate the expression of bactericidal/permeability-increasing protein (BPI), a critical antimicrobial effector, and assessed their ability to reduce the intracellular burden of mycobacteria and induce autophagy. The results demonstrate diverse immunostimulatory profiles among the lysates, highlighting differences in both inflammatory and antimicrobial responses that may be relevant for host-directed therapeutic strategies against tuberculosis. Notably, beyond the in vitro antimycobacterial activity observed for BPI, this protein was also found to be elevated in both serum and bronchoalveolar lavage fluid from patients with active TB, reflecting local and systemic immune activation. Furthermore, the reduction in BPI levels after treatment suggests its potential utility for following the dynamics of infection.

## 1. Introduction

The use of bacterial lysates as immunomodulatory agents has gained increasing attention due to their ability to enhance host defenses without eliciting a specific immune response. Bacterial lysates are complex mixtures of bacterial components generated through mechanical or chemical disruption of microbial cells [1]. They contain conserved pathogen-associated molecular patterns (PAMPs) that are recognized by pattern recognition receptors (PRRs) on immune cells, thereby triggering innate immune responses and promoting the development of adaptive immunity [2,3]. The components of bacterial lysates, such as peptidoglycan, lipoproteins, nucleic acids, and other structural elements, stimulate both mucosal and systemic immunity [4,5].

Bacterial lysates have been used for decades to prevent recurrent respiratory and urinary tract infections [1,6]. Pulmonarom has been reported to enhance dendritic cell activation and cytokine production, supporting its role in the prevention and treatment of respiratory infections [7]. Ismigen has demonstrated efficacy in reducing the number and duration of recurrent respiratory infections, as well as increasing systemic and mucosal immunoglobulins in adults [8,9,10]. Additionally, Uro-Vaxom (OM-89, OM Pharma, Geneva, CHE) is used for the prevention of recurrent urinary tract infections, with demonstrated reduced recurrence rates [11,12]. Despite widespread use, the precise cellular mechanisms by which bacterial lysates modulate human immune responses are not fully elucidated. Diverse research data indicate that bacterial lysates function as immunomodulators capable of inducing antibodies against pathogens and eliciting immunoregulatory responses in mucosal tissues [13,14]. Remarkably, they may interact with diverse immune cells, including monocytes, dendritic cells, or epithelial cells, through the engagement of surface Toll-like receptors (TLRs) with bacterial components, such as peptidoglycans or lipopolysaccharides [15]. These interactions generate the differentiation of monocytes and activate immature dendritic cells to produce specific chemokines and cytokines [1].

Macrophages play a pivotal role in orchestrating innate immune responses, particularly in the context of intracellular infections such as those caused by *M. tuberculosis* [16]. Upon activation, macrophages secrete a range of pro-inflammatory cytokines and produce antimicrobial peptides and proteins that are critical for pathogen control [17]. One such effector is bactericidal/permeability-increasing protein (BPI), a multifunctional antimicrobial protein. BPI has an antimicrobial effect that can kill Gram-negative bacteria by forming pores in their membranes, causing the loss of the proton-motive force, allowing the entry of granulysins and granzymes, and thereby favoring bacterial lysis and death. BPI facilitates phagocytosis and contributes to the processing and presentation of antigens to immune cells [18,19,20]. Recent studies have demonstrated that BPI also exhibits intracellular antimicrobial activity against *M. tuberculosis*, suggesting wider functions in host defense [21].

Gaining insights into how different bacterial lysates induce macrophage activation, cytokine production, and the expression of antimicrobial effectors, such as BPI, which can decrease intracellular bacterial growth, may reveal new strategies for host-directed therapy (HDT) in tuberculosis (TB). HDT aims to boost the host immune response, thereby enhancing pathogen clearance, reducing disease severity, and shortening treatment duration [22]. In this context, bacterial lysates could serve as a valuable adjunct to standard antibiotic treatments by priming macrophages toward a more effective antimicrobial state, potentially shortening anti-tuberculosis therapy.

This study aimed to evaluate the immunomodulatory properties of selected commercial and experimental mycobacterial lysates, including Pulmonarom, Ismigen, Uro-Vaxom, and a lysate derived from *M. tuberculosis* H37 Ra (LMtb), on human monocyte-derived macrophages (MDMs). Specifically, we investigated their cytotoxicity profiles, ability to induce pro-inflammatory cytokines (TNF-α, IL-1β, and IL-8), and their effect on BPI expression at both mRNA and protein levels. We further examined whether these lysates could modulate macrophage antimicrobial responses, including the reduction in intracellular *M. tuberculosis* burden and the activation of autophagy. Finally, BPI levels were analyzed in bronchoalveolar lavage (BAL) and serum samples from TB patients to determine their relevance as potential diagnostic biomarkers of infection.

## 2. Materials and Methods

### 2.1. Human Samples

This protocol was approved by the Review Boards at the Instituto Nacional de Enfermedades Respiratorias “Ismael Cosio Villegas” (INER) in Mexico City. Consent for the use of buffy coats from the blood bank for the isolation of monocytes and residual clinical specimens from bronchoalveolar lavage of patients diagnosed with pulmonary TB was waived by the review board, provided the identity of the subjects remained anonymous. Informed consent was procured from healthy and TB donors before the collection of serum, in accordance with the Declaration of Helsinki. BALs were centrifuged to remove cellular debris and then filtered through a 0.22 μm membrane to eliminate potential microbial contaminants. Serum was obtained from venous blood by centrifugation. All samples were aliquoted and stored at −70 °C until analysis.

### 2.2. Bacterial Strains and Culture Conditions

The avirulent *M. tuberculosis* H37 Ra and the virulent H37 Rv strains were obtained from the American Type Culture Collection (ATCC 25177 and 27294, Rockville, MD, USA). H37 Ra was transformed with the pCherry8 plasmid (Addgene #24663, Cambridge, MA, USA), which encodes the fluorescent reporter *mCherry* [23]. Bacterial stocks were stored at −70 °C and cultured in Middlebrook 7H9 broth supplemented with 10% ADC (Becton Dickinson, San Jose, CA, USA) and 0.2% glycerol. Colony-forming units (CFUs) were enumerated on Middlebrook 7H10 agar supplemented with 10% OADC and 0.05% glycerol (7H10c), incubated at 37 °C for 21 days.

Prior to macrophage infection, bacterial suspensions were prepared by thawing and centrifuging the stocks (8000 rpm, 5 min), washing with antibiotic-free RPMI (Lonza, Walkersville, MD, USA) supplemented with 200 mM L-glutamine (Lonza) and 10% activated human serum (Valley Biomedical, Winchester, VA, USA) and disaggregating using 5 mm glass beads vortexed for 5 min. The suspension was then centrifuged at 800 rpm for 2 min to remove aggregates, as previously described [24].

### 2.3. Cell Culture and Differentiation

Peripheral blood mononuclear cells (PBMCs) were isolated from the buffy coats of healthy blood donors using density gradient centrifugation using Lymphoprep. Monocytes were subsequently enriched by magnetic positive selection (Miltenyi Biotec, Auburn, CA, USA). Monocytes were cultured for 7 days to generate monocyte-derived macrophages (MDMs) in RPMI-1640 medium supplemented with 200 mM L-glutamine (Lonza) and 10% heat-inactivated human serum (Valley Biomedical, Winchester, VA, USA). Cultures were maintained at 37 °C in a humidified atmosphere containing 5% CO_2_. Cell viability was assessed using trypan blue exclusion in a Neubauer chamber.

### 2.4. Bacterial Lysates Working Concentrations

We analyzed various commercial bacterial lysates: Pulmonarom (Sanofi Winthrop Industrie, Gentilly, FRA), Urovaxom (OM-89, OM Pharma, Geneva, CHE), and Ismigen (Lallemand Pharma International, Zug, CHE). Their composition is described in Appendix A. We included an in-house-made lysate of *M. tuberculosis* H37Ra, prepared from a culture grown to the stationary phase. Briefly, heat-killed *M. tuberculosis* was prepared by incubating the bacteria at 80 °C for 50 min, followed by vortexing for 10 min with 5 glass beads. The lysate was obtained by sonicating twice at 80 cycles for 30 sec each using an Ultrasonik 28× sonicator (NEY, Yucalpa, CA, USA). Complete bacterial killing was confirmed by the absence of growth in 7H10 agar plates.

The monocytes were seeded at 3 × 10^5^ cells/well in 48-well plates in culture medium and incubated for 7 days. Then, bacterial lysates were added to the cells and incubated for 72 h. Cell viability was assessed every 24 h using the MTT reduction assay. Results are reported as percentage viability relative to the untreated control (considered as 100%). The working concentrations were selected within the non-cytotoxicity range (Appendix A).

### 2.5. Effect of Bacterial Lysates on BPI Expression

#### 2.5.1. mRNA Expression

MDMs were stimulated with bacterial lysates for 72 h. At 24, 48, and 72 h, total RNA was extracted using the RNeasy Mini Kit (QIAGEN, Hilden, Germany) according to the manufacturer’s instructions. Complementary DNA (cDNA) was synthesized from 1 µg of total RNA using the SuperScript™ Master Mix (Invitrogen, Carlsbad, CA, USA). Quantitative PCR was performed with TaqMan assays for BPI-FAM (Hs01552756_m1, Applied Biosystems, Foster City, CA, USA) and 18S ribosomal RNA-VIC (Applied Biosystems) as the endogenous control. Relative BPI expression was calculated using the 2^−ΔΔCt^ method, with cells cultured in medium alone serving as the calibrator, and results were expressed as fold change.

#### 2.5.2. Western Blot

To evaluate BPI protein expression, we incubated 3 × 10^6^ macrophages per well with bacterial lysates at varying concentrations for 72 h, and protein expression was assessed every 24 h. Total proteins were extracted and quantified using a Bio-Rad Protein Assay (Bio-Rad Laboratories, Hercules, CA, USA). Equal amounts of protein were separated by SDS-PAGE and transferred to PVDF membranes. The membrane was blocked with 5% non-fat milk at room temperature overnight and then incubated with anti-human BPI antibody (Monoclonal Mouse IgG_2B_, R&D Systems, Minneapolis, MN, USA) for 2 h. After washing, membranes were incubated with an HRP-labeled secondary antibody at room temperature. Protein bands were visualized using Luminol (Bio-Rad), and densitometric analysis was performed using Image Lab v5 (Bio-Rad). Endogenous BPI was detected at approximately 52 kDa.

#### 2.5.3. Fluorescence Microscopy

Macrophages (0.175 × 10^6^ cells/well) were seeded in Lab-Tek II chamber slides (Thermo Fisher Scientific, Waltham, MA, USA) and stimulated with bacterial lysates for 48 h. They were then infected or not with avirulent mycobacteria that had been previously opsonized using 10% active human serum at a multiplicity of infection of 10 in antibiotic-free RPMI medium. The cells were incubated for 1 h at 37 °C in a 5% CO_2_ atmosphere, followed by thorough washing to eliminate extracellular bacteria. Immediately after infection, cells were re-stimulated with freshly prepared bacterial lysates diluted in RPMI medium supplemented with 10% heat-inactivated human serum at the indicated concentrations. Cultures were maintained at 37 °C and 5% CO_2_ for 24 h. Finally, cells were fixed with 4% paraformaldehyde.

For BPI detection, cells were incubated with an anti-hBPI mouse antibody (R&D Systems, Minneapolis, MN, USA) and an anti-mouse antibody coupled with Alexa Fluor 488 rabbit antibody (Thermo Fisher Scientific, Waltham, MA, USA). Nuclei were counterstained with Hoechst dye (Enzo Life Sciences, Farmingdale, NY, USA). Imaging was performed using an AxioScope.A1 fluorescence microscope (Carl Zeiss, Oberkochen, Germany), and data were processed using ZEN Pro 2012 software (Carl Zeiss).

### 2.6. Immunomodulatory Effect of Bacterial Lysates on Human Macrophages

#### 2.6.1. Modulation of Cytokine Response (TNF-α, IL-1β, and IL-8)

MDMs were stimulated with bacterial lysates for 24 h, and culture supernatants were collected and stored at −20 °C until further analysis. Cytokine concentrations were determined by enzyme-linked immunosorbent assay (ELISA). TNF-α levels were quantified as previously described [25], while IL-1β and IL-8 were measured using ELISA Flex kits (Mabtech, Stockholm, Sweden), according to the manufacturer’s instructions. Absorbance was read on an HT Multi-Mode Microplate Reader (Biotek, WA, USA) at 450 nm. The results are presented as the mean value for duplicate wells in six independent experiments.

#### 2.6.2. Modulation of Macrophage Antimicrobial Responses

To investigate the immunomodulatory properties of bacterial lysates, we assessed their effect on the antimicrobial activity of macrophages against intracellular *M. tuberculosis* H37Rv. MDMs (0.2 × 10^6^ cells per well in 96-well plates) were stimulated with bacterial lysates for 48 h and then infected with mycobacteria at a multiplicity of infection of 1 or 5 in antibiotic-free RPMI medium. The infected cells were incubated for 1 h at 37 °C in a 5 % CO_2_ atmosphere, followed by thorough washing to eliminate extracellular bacteria. Immediately after infection, cells were re-stimulated with freshly prepared bacterial lysates diluted in RPMI medium containing 10% heat-inactivated human serum at the specified concentrations. The cultures were maintained at 37 °C and 5% CO_2_ for three days. At the end of the incubation period, intracellular bacterial loads were determined. MDMs were lysed with 0.1% SDS, followed by neutralization using a 10% BSA solution, and viable intracellular bacteria were quantified by CFU enumeration.

#### 2.6.3. Autophagy Induction by Bacterial Lysates

Macrophages were seeded in a Lab-Tek II chamber and then treated and infected as described above, and incubated for 24 h. Cells were stained with rabbit anti-LC3B conjugate with Alexa Fluor 488 (Novus Biologicals, CO, USA) as a marker of autophagosome and counterstained with Hoechst to detect nuclei. Cells were visualized using an AxioScope.A1 fluorescence microscope.

### 2.7. Detection of BPI in Serum and Bronchoalveolar Lavage Fluid in TB Patients

BPI concentrations were measured in serum and bronchoalveolar lavage of patients with pulmonary TB, and serum of healthy donors, using a commercially available ELISA kit (Hycult Biotech, Uden, The Netherlands), following the manufacturer’s instructions. Prior to analysis, the samples were thawed and added to ELISA plates that were pre-coated with anti-BPI capture antibodies. After incubation and washing steps, a biotinylated detection antibody and streptavidin-HRP conjugate were applied sequentially. The enzymatic reaction was developed with TMB substrate and stopped with sulfuric acid. Absorbance was read at 450 nm using a microplate reader, and BPI levels were determined from a standard curve generated with recombinant BPI.

### 2.8. Statistical Analysis

All statistical analyses were performed using non-parametric methods and analyzed with GraphPad version 10.6.0 (GraphPad Software, San Diego, CA, USA). Overall differences among experimental conditions were assessed using the Friedman test with Dunn’s post hoc correction for multiple comparisons. Individual conditions were compared either against a fixed reference value using the Wilcoxon signed-rank test, or against the uninfected control using the Mann-Whitney U test. Statistical significance was defined as *p* < 0.05.

## 3. Results

To determine the optimal concentrations for functional assays, we first evaluated the cytotoxic effects of bacterial lysates on MDMs at 24, 48, and 72 h (Appendix A). Pulmonarom exhibited minimal cytotoxicity, while Ismigen, Uro-Vaxom, and LMtb showed dose- and time-dependent effects. No cytotoxicity was detected after 72 h with Pulmonarom at 1.25–10 × 10^6^ cells/mL, Ismigen at 6–12.5 μg/mL, Uro-Vaxom at 6–25 μg/mL, or LMtb at 1–2.5 × 10^6^ cells/mL. These maximum non-cytotoxic concentrations were subsequently used to stimulate MDMs to investigate their effects on BPI expression, cytokine secretion, and intracellular survival of *M. tuberculosis.*

### 3.1. Bacterial Lysates Induce Cytokine-Mediated Activation in Human Macrophages

MDMs were stimulated with different commercial bacterial lysates or LMtb for 24 h. The concentration of TNF-α, IL-1β, and IL-8 in the culture supernatants was quantified by ELISA (Figure 1). Stimulation with Ismigen led to a significant increase in TNF-α, IL-1β, and IL-8 production (mean: 7290 ± SEM: 395 pg/mL, 2569 ± 464 pg/mL, and 335 ± 28 ng/mL, respectively) compared to the control. LMtb also induced a significant production of TNF-α, IL-1β, and IL-8 (mean: 8543 ± 2097 pg/mL, 1379 ± 158 pg/mL, and 441 ± 49 ng/mL, respectively) compared to the untreated control.

In contrast, Pulmonarom and Uro-Vaxom induced modest increases in TNF-α production, which were not statistically significant (945 ± 395 pg/mL, and 643 ± 249 pg/mL, respectively), and did not induce IL-1β or IL-8 production. Overall, these findings indicate that Ismigen and LMtb are potent inducers of pro-inflammatory cytokines in human macrophages. At the same time, Pulmonarom and Uro-Vaxom exhibit weaker immunostimulatory effects, highlighting differential immunostimulatory properties among the tested bacterial lysates.

### 3.2. Bacterial Lysates Upregulate BPI in Human Macrophages

To assess the effect of bacterial lysates on BPI expression, MDMs were stimulated with Pulmonarom, Ismigen, Uro-Vaxom, or LMtb for 24, 48, and 72 h (Figure 2). All bacterial lysates significantly upregulated BPI mRNA expression at 24 h, with mean fold changes ranging from 2.6 to 8.7 compared to unstimulated MDMs (control) (*p* < 0.05). The strongest induction was observed in cells stimulated with Ismigen and Pulmonarom. BPI transcript levels returned to baseline at 48 and 72 h, with no statistically significant differences observed at these later time points. These results indicate that bacterial lysates elicit a rapid but transient transcriptional upregulation of BPI in human macrophages.

To evaluate whether this transcriptional response was reflected at the protein level, BPI expression was assessed by Western blot over time. As shown in Figure 3, a slight but non-significant increase in BPI expression levels was observed at 24 h after stimulation with all lysates (mean fold changes ranging from 0.9 to 1.1). At 48 h, a significant induction of BPI protein was detected in MDMs treated with Pulmonarom (2.2 ± 0.6 fold), Ismigen (1.6 ± 0.5 fold), Uro-Vaxom (1.5 ± 0.2 fold), and LMtb (1.6 ± 0.1 fold) (*p* < 0.05). By 72 h, only Pulmonarom-treated cells maintained significantly elevated BPI expression (1.5 ± 0.2 fold), while the other treatments showed a non-significant trend toward increased expression. These data confirm that bacterial lysates induce a time-dependent but transient increase in BPI expression at both the transcript and protein levels, with Pulmonarom demonstrating the most sustained effect.

Further, we determined the effect of *M. tuberculosis* infection on BPI expression at the cellular level. MDMs were infected 48 h post-stimulation with the lysates. Fluorescence microscopy imaging revealed that cells treated with all the lysates expressed BPI in a small dotted pattern (Figure 4). The percentages of BPI-positive cells increased significantly, reaching 29 ± 1.0%, 21 ± 4.0%, 16 ± 1.2%, and 20 ± 1.2% with Pulmonarom, Ismigen, Uro-Vaxom, and LMtb, respectively. Notably, subsequent infection with *M. tuberculosis* maintained high BPI expression, with values of 34 ± 2.9%, 31 ± 3.1%, 14 ± 1.5% and 23 ± 2.7%, for Pulmonarom, Ismigen, Uro-Vaxom, and LMtb, respectively (Figure 5). These findings suggest that mycobacterial infection does not interfere with lysate-induced BPI expression.

### 3.3. Bacterial Lysates Contributed to Inhibiting the Intracellular Growth of M. tuberculosis in MDMs, Inducing Autophagy

Pre-treatment with bacterial lysates significantly induced an antibacterial state of the macrophages. The intracellular burden of *M. tuberculosis* in MDMs after 3 days of incubation was compared to untreated controls (Figure 6). In MDMs infected at a multiplicity of infection of 1, intracellular bacterial loads were reduced by 56%, 70%, 67%, and 69% following treatment with Pulmonarom, Ismigen, Uro-Vaxom, and LMtb, respectively. Under a higher infection burden (MOI 5), bacterial reductions were 46%, 72%, 70%, and 58% for the same treatments.

We further investigated the capacity of the bacterial lysates to induce autophagy as a potential mechanism for reducing intracellular mycobacterial burden. All bacterial lysates promoted autophagosome formation, exhibiting vesicle formation unlike the untreated control, and in a manner comparable to chloroquine, a well-established autophagy inducer (Figure 7). The percentages of LC3-positive cells increased significantly, by 23%, 21%, and 20% with chloroquine, Pulmonarom, and Ismigen, respectively, and by 8% and 12% with Uro-Vaxom and *M. tuberculosis* lysate (Figure 7B).

Recombinant human BPI also induced autophagosome formation (Appendix A), reaching levels comparable to those observed with chloroquine. In contrast, MDMs treated with Pulmonarom and infected with *M. tuberculosis* showed a significant reduction in autophagosome formation, from 21% to 7%, indicating that mycobacteria act as potent inhibitors of the autophagic process (Figure 8).

### 3.4. BPI Is Increased in Serum and BAL During Active TB

Given the potential role of BPI in clearing infection, we investigated the BPI levels in the serum of patients with active TB and healthy individuals. As shown in Figure 9, serum concentrations of BPI were higher in TB patients (mean: 39.6 ng/mL ± SEM: 4.2 ng/mL) than those observed in healthy controls (20.3 ± 2.3 ng/mL) (*p* < 0.05), suggesting an upregulation of this antimicrobial protein during active infection. To further determine the compartmentalization of BPI in TB, we quantified its levels in BAL fluid. BAL concentrations of BPI were significantly higher (1061 ± 268 ng/mL) than serum (*p* < 0.05), indicating a localized enrichment of BPI at the site of infection. Moreover, paired analysis of serum samples from TB patients at treatment initiation (t1; 0–7 days) and after 60 days of anti-TB therapy (t2) showed a significant decrease in BPI concentrations during the early phase of treatment (*p* < 0.05). This reduction suggests that BPI levels may reflect the resolution of pulmonary infection and inflammation following effective antimicrobial therapy.

## 4. Discussion

Understanding the immunomodulatory effects of bacterial lysates is of growing interest due to their potential to enhance host defense mechanisms without directly targeting pathogens. Unlike traditional antibiotics, bacterial lysates act by priming the innate immune system, promoting cytokine release, and upregulating antimicrobial proteins, which may improve the control of intracellular infections such as TB. Investigating how different lysates affect human macrophage responses can provide valuable insights for developing host-directed therapies that boost immune function and improve treatment outcomes in infectious diseases.

We observed that bacterial lysates induce distinct cytokine profiles, reflecting differential engagement of pattern recognition receptors (PRRs). Pulmonarom and Uro-Vaxom slightly induced IL-1β, IL-8, and TNF-α production. Pulmonarom has been reported to increase the expression of HLA-DR molecules and TLR2, TLR3, TLR6, and TLR7 receptors, accompanied by the increased production of cytokines IL-8, MCP-1, IL-6, and IL-4 in monocyte-derived dendritic cells [7]. However, it is possible that the lower responses to Pulmonarom and Uro-Vaxom may arise from reduced amounts of lipoproteins/LPS, which serve as TLR ligands, or partial depletion of nucleic acid/DNA motifs that stimulate endosomal TLR9/NOD-like receptors during alkaline extraction [1,7]. Alternatively, these bacterial lysates may induce cytokines not measured in this study.

Ismigen and LMtb were the most potent cytokine inducers. Their cytokine pattern is consistent with robust TLR2/TLR4 engagement, leading to canonical activation via NF-κB [26]. This finding aligns with previous studies demonstrating that commercial lysates, such as OM-85V, and proteins from bacterial lysates induce inflammatory cytokines in murine cells [27] and in the human myeloid cell line THP-1 [1,28].

From a host defense perspective, the induction of TNF-α and IL-1β is relevant because both cytokines are essential for maintaining granuloma integrity and the nitric oxide-dependent control of *M. tuberculosis* in human macrophages [29]. IL-8 production further suggests that Ismigen and LMtb can recruit neutrophils to infection sites, potentially amplifying bactericidal activity [30].

We observed that bacterial lysates induce a rapid yet transient upregulation of BPI, emphasizing an early antimicrobial mechanism. All lysates triggered an early (24 h) surge in BPI mRNA; however, only Pulmonarom sustained protein expression up to 72 h. Although BPI is best known for neutralizing Gram-negative LPS, recent work demonstrates that recombinant human BPI can penetrate macrophages and restrict the intracellular growth of *M. tuberculosis* [21]. Our results extend these findings, demonstrating that endogenous BPI expression can be upregulated by non-mycobacterial lysates and may contribute to antimycobacterial defense. Notably, BPI expression was maintained in infected macrophages, further supporting this essential role during infection.

All four lysates reduced the *Mycobacterium* burden by 46–72 % after 72 h. The most substantial reductions paralleled the highest TNF α/IL-1β outputs (Ismigen and LMtb), underscoring the importance of classic macrophage-activating cytokines for controlling bacterial replication. Furthermore, despite relatively modest cytokine induction, Pulmonarom decreased the intracellular bacterial burden to 56% supporting a complementary role for sustained BPI expression. Exogenous BPI alone curtails intracellular mycobacteria [21], these results suggest that Pulmonarom’s prolonged BPI upregulation compensates for its lower pro-inflammatory drive. Notably, the effect of bacterial lysates-induced innate immunity increases the opportunity for host-directed adjunct therapy to shorten antibiotic treatment and lower the risk of relapse.

Pulmonarom and Ismigen visibly stimulated autophagosome formation relative to controls, whereas Uro-Vaxom and LMtb elicit a low response. This differentiation suggests that specific lysate components may engage pathogen-associated molecular patterns (PAMPs) or damage-associated molecular patterns (DAMPs), triggering xenophagy in macrophages. The finding that recombinant BPI promotes autophagy further supports the notion that defined molecular triggers can enhance autophagic responses. Interestingly, when MDMs were infected with *M. tuberculosis*, autophagy levels remained largely sustained under lysate treatment, except for Pulmonarom, which was associated with a significant reduction in autophagosome formation. Such a decrease is consistent with the action of *M. tuberculosis* virulence factors, such as LprE or ESX1 effectors, which suppress LC3 conversion or autophagic flux [31,32]. These observations suggest that, while bacterial lysates generally promote autophagy, Pulmonarom may interact differently during infection, potentially revealing specific host–pathogen–lysate dynamics that warrant further mechanistic investigation [7,14].

In line with our in vitro results, BPI measurement in patients supports the notion that it may contribute in vivo to the control of TB. BPI levels were significantly higher in serum from TB patients compared to healthy individuals, supporting the systemic upregulation of BPI during active infection. Notably, BPI was further enriched in BALs, suggesting that BPI is not only systemically elevated during TB but is also concentrated within the lung environment, potentially contributing to the host immune response against *M. tuberculosis*. Its accumulation in BAL may reflect alveolar macrophage activation and innate immune engagement during the early stages of TB pathogenesis. We determined that after 60 days of anti-TB treatment, the BPI levels significantly decreased. These findings indicate that the BPI protein has a potential role in bacterial burden control and resolution of inflammation in active pulmonary TB, and the dynamics of its production and regulation warrant further investigation.

## 5. Conclusions

This study underscores the immunomodulatory potential of bacterial lysates in human macrophages and their relevance as host-directed therapeutic candidates for TB. The bacterial lysates displayed distinct profiles in terms of cytokine induction and BPI upregulation. Ismigen and LMtb elicited a strong yet transient inflammatory response, resulting in the rapid intracellular killing of *M. tuberculosis*. In contrast, Pulmonarom induced a weaker cytokine response but sustained BPI expression, achieving comparable bacterial control. Uro-Vaxom also enhanced macrophage activation and bacterial clearance. The role of BPI in controlling infection may be supported by its decline with anti-TB treatment in patients, when bacterial burdens are lower. These findings suggest that the differential immunostimulatory properties of bacterial lysates could be strategically leveraged in the design of host-directed therapies in combination with conventional antimicrobials and warrant further evaluation in preclinical models of infection.

## Figures and Tables

**Figure 1 biomolecules-15-01446-f001:**
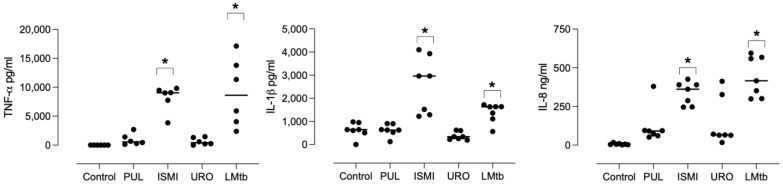
Pro-inflammatory cytokine production by macrophages in response to bacterial lysates. MDMs from healthy donors were stimulated with bacterial lysates: Pulmonarom (10 × 10^6^ cells/mL; PUL), Ismigen (12.5 μg/mL; ISMI), Uro-Vaxom (25 μg/mL; URO), or *M. tuberculosis* lysate (2.5 × 10^6^ cells/mL; LMtb). After 24 h, levels of TNF-α, IL-1β, and IL-8 in the culture supernatants were quantified by ELISA. Data are presented as mean ± SEM (*n* = 7). Data were analyzed using the Friedman test followed by Dunn´s multiple comparisons test. * *p* < 0.05 compared with the control.

**Figure 2 biomolecules-15-01446-f002:**
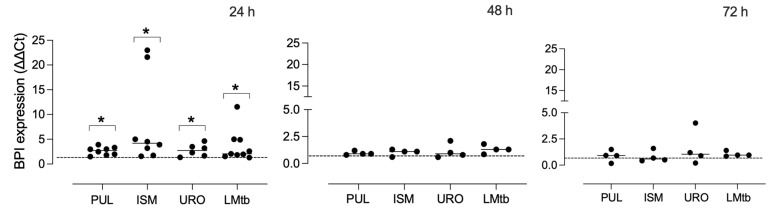
Human BPI mRNA expression in MDMs stimulated with commercial bacterial lysates. MDMs were stimulated with commercial bacterial lysates: Pulmonarom (10 × 10^6^ cells/mL; PUL), Ismigen (12.5 μg/mL; ISM), Uro-Vaxom (25 μg/mL; URO), or *M. tuberculosis* lysate (2.5 × 10^6^ cells/mL; LMtb). Total RNA was extracted at 24, 48, and 72 h post-stimulation, and BPI gene expression was quantified by RT-qPCR. Expression levels were normalized to bacterial 16S rRNA and analyzed using the ΔΔCt method. Each dot represents an individual donor (*n* = 4–9) and the dashed horizontal line indicates the value of 1, corresponding to unstimulated control. Data were analyzed using the Wilcoxon signed-rank test against a constant value of 1; * *p* < 0.05.

**Figure 3 biomolecules-15-01446-f003:**
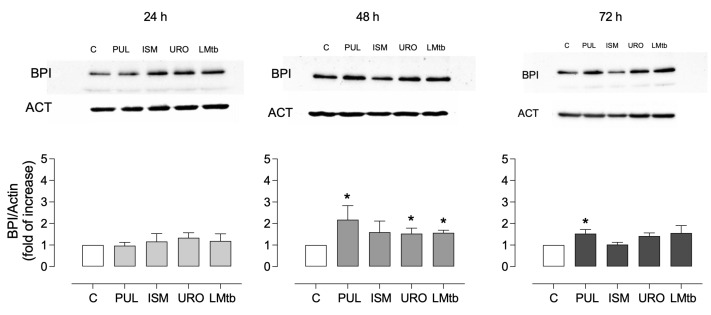
BPI protein expression in macrophages stimulated with commercial bacterial lysates. MDMs were treated with Pulmonarom (10 × 10^6^ cells/mL; PUL), Ismigen (12.5 μg/mL; ISMI), Uro-Vaxom (25 μg/mL; URO), or *M. tuberculosis* lysate (2.5 × 10^6^ cells/mL; LMtb) for 24, 48, and 72 h; untreated cells were used as controls (C). BPI expression was assessed by Western blot with actin (ACT) as a loading control (original images can be found in Appendix A). Densitometric analysis (bottom panels) shows BPI levels, normalized to actin, and expressed as a fold change relative to the controls. Data were analyzed using the Wilcoxon signed-rank test against the constant value of 1 (n = 4–6). * *p* < 0.05.

**Figure 4 biomolecules-15-01446-f004:**
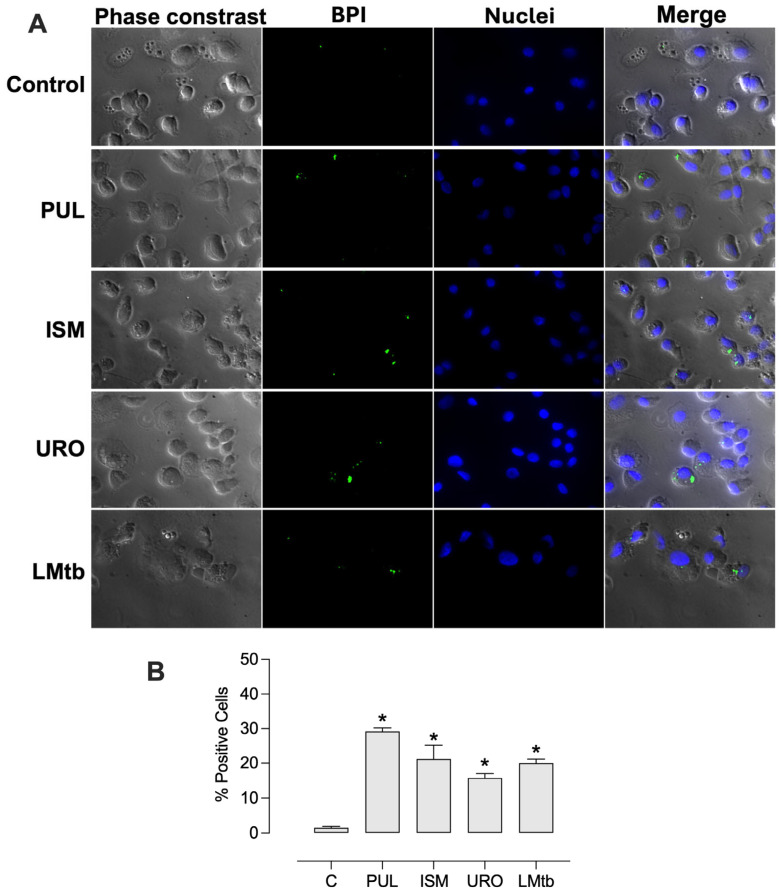
BPI expression in human macrophages treated with bacterial lysates. MDMs untreated (control) or treated with Pulmonarom (10 × 10^6^ cells/mL; PUL), Ismigen (12.5 μg/mL; ISm), Uro-Vaxom (25 μg/mL; URO), or *M. tuberculosis* lysate (2.5 × 10^6^ cells/mL; LMtb) for 48 h and then BPI was detected via fluorescence microscopy using an anti-human BPI antibody and a secondary antibody coupled with Alexa Fluor 488 and nuclei were stained with Hoechst. Images are representative of six independent experiments; images were acquired at 100× (**A**). The percentage of BPI-positive cells was assessed by counting 8- 10 fields per condition (**B**). Data were analyzed using the Friedman test followed by Dunn’s multiple comparisons test; * *p* < 0.05.

**Figure 5 biomolecules-15-01446-f005:**
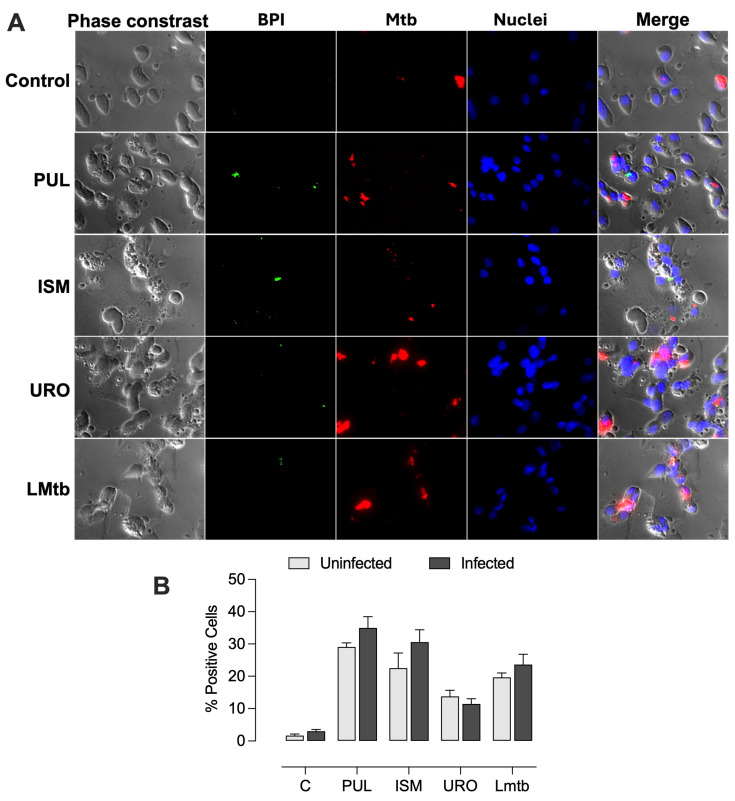
BPI expression in infected macrophages treated with bacterial lysates. MDMs were treated with Pulmonarom (10 × 10^6^ cells/mL; PUL), Ismigen (12.5 μg/mL; ISM), Uro-Vaxom (25 μg/mL; URO), or *M. tuberculosis* lysate (2.5 × 10^6^ cells/mL; LMtb) for 48 h and then infected with H37Ra. BPI was detected by fluorescence microscopy using an anti-human BPI antibody and Alexa Fluor 488-conjugated secondary antibody; mycobacteria express mCherry, and nuclei were stained with Hoechst. Images are representative of six independent experiments (100×, **A**). The percentage of BPI-positive cells was determined by counting 8–10 fields per condition (**B**). Data were analyzed using the Wilcoxon signed-rank test comparing each treatment to its uninfected control (C).

**Figure 6 biomolecules-15-01446-f006:**
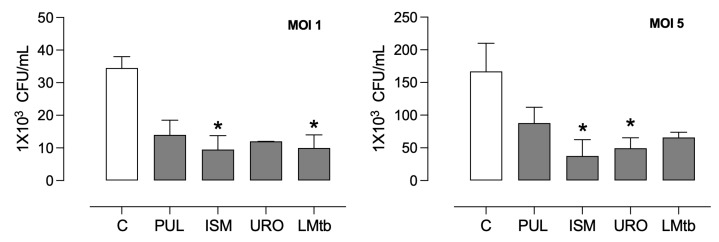
Inhibitory effect of bacterial lysates on the intracellular growth of *M. tuberculosis*. MDMs were pre-treated with Pulmonarom (10 × 10^6^ cells/mL; PUL), Ismigen (12.5 μg/mL; ISM), Uro-Vaxom (25 μg/mL; URO), or *M. tuberculosis* lysate (2.5 × 10^6^ cells/mL; LMtb) for 48 h, then infected with virulent *M. tuberculosis* strains for 3 days. Cells were lysed, and intracellular bacteria were serially diluted and plated on 7H10 medium; CFUs were counted after 21 days. Data were analyzed using the Friedman test with Dunn’s multiple comparisons test. **p* < 0.05 versus untreated cells (C; *n* = 4).

**Figure 7 biomolecules-15-01446-f007:**
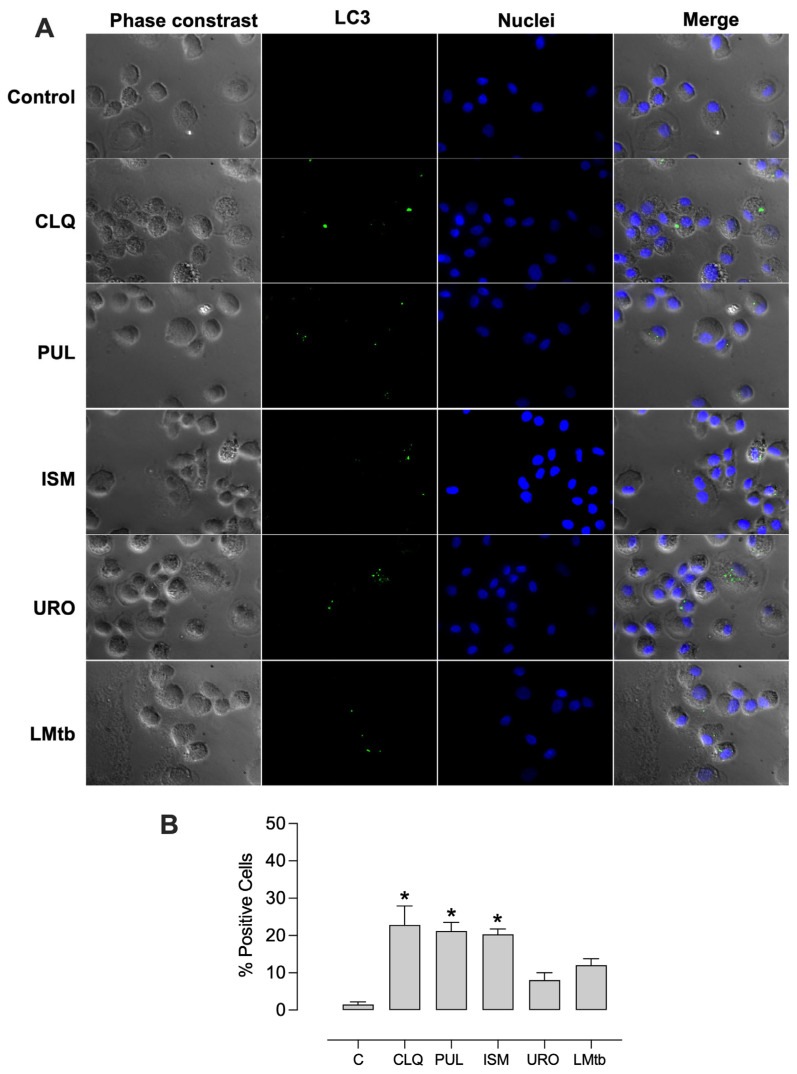
Bacterial lysates induced autophagy in macrophages. MDMs were treated with Pulmonarom (10 × 10^6^ cells/mL; PUL), Ismigen (12.5 μg/mL; ISM), Uro-Vaxom (25 μg/mL; URO), *M. tuberculosis* lysate (2.5 × 10^6^ cells/mL; LMtb), Chloroquine (CLQ), or left untreated (C) for 48 h. Autophagosomes were detected by fluorescence microscopy using an anti-human LC3-FITC antibody and Hoechst-stained nuclei; merged images show autophagy-positive cells. Images are representative of five independent experiments (**A**), and the percentage of positive cells was determined by counting 8–10 fields per condition (**B**). Data were analyzed using the Friedman test followed by Dunn’s multiple comparisons test; **p* < 0.05.

**Figure 8 biomolecules-15-01446-f008:**
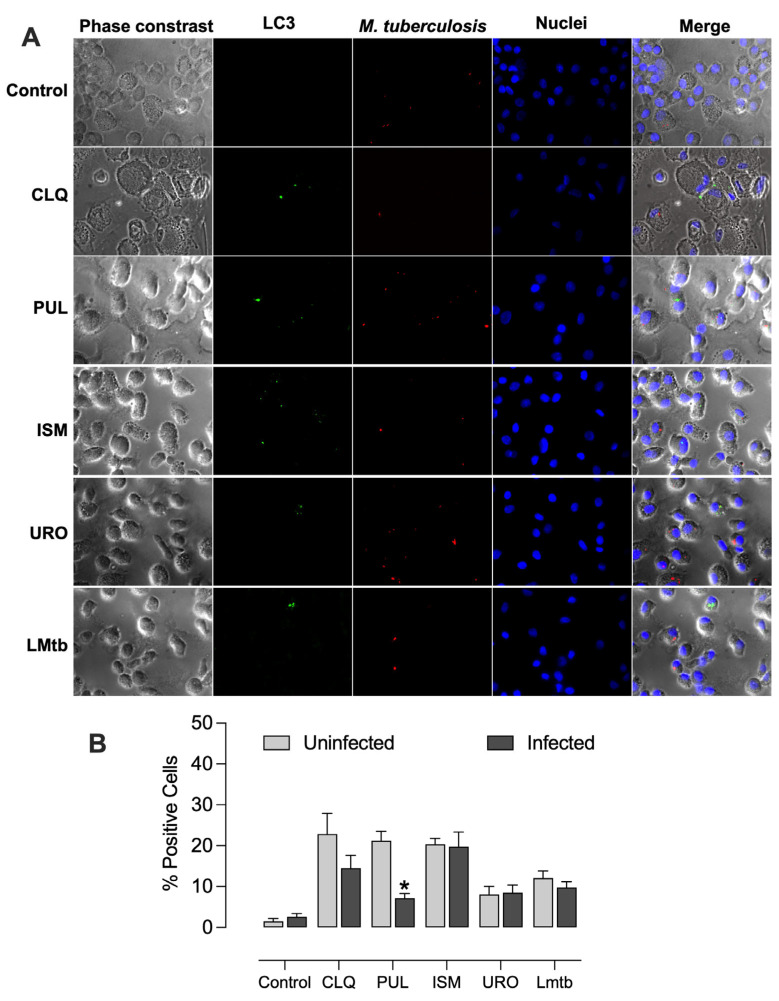
**Bacterial lysates induced autophagy in infected macrophages.** MDMs were treated with Pulmonarom (10 × 10^6^ cells/mL; PUL), Ismigen (12.5 μg/mL; ISM), Uro-Vaxom (25 μg/mL; URO), *M. tuberculosis* lysate (2.5 × 10^6^ cells/mL; LMtb), or left untreated (C) for 48 h and then infected with *M. tuberculosis*. Autophagosomes were detected by fluorescence microscopy using an anti-human LC3-FITC antibody; mycobacteria express mCherry, and nuclei were stained with Hoechst. Images are representative of five independent donors (**A**), and the percentage of LC3-positive cells was determined by counting 8–10 fields per condition (**B**). Data were analyzed using the Wilcoxon signed-rank test comparing each treatment with the uninfected control; **p* < 0.05.

**Figure 9 biomolecules-15-01446-f009:**
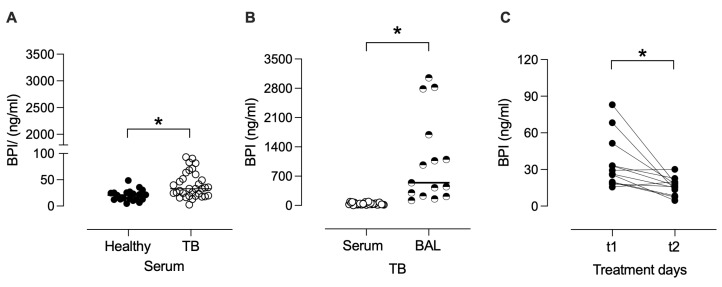
BPI concentrations in serum and bronchoalveolar lavage (BAL) during active TB. BPI levels were measured by ELISA in serum from healthy donors (black circles) and TB patients (open circles), and in BAL fluid from TB patients (half-filled circles). Paired analysis shows serum BPI concentrations in TB patients at treatment initiation (t1; 0–7 days) and after 60 days of anti-TB therapy (t2). Data are presented as individual values with medians (n = 12–35). Statistical analyses were performed using the Mann–Whitney test (**A**,**B**) and the Wilcoxon signed-rank test (**C**); **p* < 0.05.

## Data Availability

The data that support the findings of this study are available upon reasonable request from the corresponding author.

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
