# Peer review of "Bacterial Lysates Modulate Human Macrophage Responses by Inducing BPI Production and Autophagy"

_biomolecules, 2025, doi:10.3390/biom15101446_

Round 1
Reviewer 1 Report
Comments and Suggestions for Authors
The manuscript demonstrates the effect of commercial bacterial lysates on cytokine production and control of M. tuberculosis infection in in vitro culture of human macrophages, in addition to evaluating BLP production and its role during in vitro infection with M. tuberculosis. Finally, the authors demonstrate the BLP expression profile in serum and bronchial lavage fluid of tuberculosis patients before and after treatment, demonstrating a certain correlation between the in vitro findings and in vivo findings in the patient. Despite the quality of the data presented, the authors present results that are not interpretable. The authors interpret fluorescence images in Figures 4, 5, and 7, indicating differences between the groups. However, the differences cannot be seen in the images presented. The authors should present quantitative data (% of labeled cells, fluorescence intensity per field, etc.) and perform analysis to verify whether there is a true difference between the groups.
Author Response
We appreciate the reviewer’s comment regarding the interpretability of the fluorescence images in Figures 4, 5, and 7. In response, we have improved the quality of these figures by increasing their resolution (dpi) to enhance clarity. Moreover, we quantified the percentage of BPI-positive cells (Figures 4 and 6) and LC3-positive cells, a marker of autophagy (Figures 7 and 8). In each case, the percentage of positive cells was assessed by counting 8- 10 fields per condition. These additions allow us to present quantitative data whose statistical analysis corroborated the significant changes more clearly in the relevant data.
1) All bacterial lysates significantly increased the expression of BPI
2) Treatment with bacterial lysates increased the proportion of LC3-positive cells, although statistical significance was achieved only with Pulmonarom and Ismigen. Interestingly, when cells pretreated with lysates were subsequently infected with M. tuberculosis, Pulmonarom induced a significant reduction in LC3-positive cells (autophagy), while the other lysates showed no significant effect.
The new data added to the manuscript reinforce the importance of BPI during infection and provide more unmistakable evidence supporting the role of bacterial lysates in modulating host responses. These revisions aim to enhance the clarity, interpretability, and overall strength of our findings, aligning with the reviewer’s valuable suggestions.

Reviewer 2 Report
Comments and Suggestions for Authors
The manuscript explores the immunomodulatory properties of different bacterial lysates in human macrophages, with a particular focus on their ability to restrict M. tuberculosis growth, induce cytokine responses, and stimulate autophagy. The clinical data on bactericidal/permeability-increasing protein (BPI) levels in serum and bronchoalveolar lavage (BAL) fluid of tuberculosis (TB) patients adds value and increases the biomedical relevance. The manuscript is generally well-structured, figures are relevant and well-described. The use of both macrophage models and patient samples strengthens the conclusions. Overall, the manuscript makes a meaningful contribution to the understanding of how bacterial lysates modulate human macrophage responses and how BPI may serve as both a biomarker and an effector molecule in TB. I believe that the manuscript may be published after substantial clarifications and corrections.
Major Comments:
Lines 249, 251. “335 ±28 µg/mL”, “441 ±49 µg/mL” ─ The unit µg/mL is most likely an error, since the concentration is unusually high. Perhaps pg/mL is intended?
Figure 1. The IL-8 concentration on the ordinate axis may contain a unit error (µg/mL). It is likely meant to be pg/mL.
Lines 388, 389, 392. Serum and BAL BPI values. Missing units makes comparison meaningless. It is unclear whether values are pg/mL, ng/mL, or μg/mL. This is a major factual omission.
Figure 9. The ordinate axis shows concentrations in µg/mL. Some “half-filled circles” indicate BPI levels of about 2800 µg/mL, which appears unusually high. Please verify the units.
Minor Comments:
Line 89. “Isamel Cosio Villegas” ─ Please check, this may be a typographical error.
Line 128. “Pulmonaron” ─ Please check the spelling; earlier and later in the text it appears as “ Pulmonarom”.
Line 136. “The MNs were seeded” ─ Does this refer specifically to monocytes (MNs) or already to monocyte-derived macrophages (MDMs)? Please clarify.
Lines 161-162. “anti-human BPI antibody (R&D Systems, Minneapolis, MN, USA)” ─ Please provide more detailed information about the antibody (monoclonal, mouse?).
Lines 171. “human serum” ─ Please specify the concentration..
Lines 206. “using 10% BSA” ─ “using 10% BSA solution”.
Lines 229-231. “For nonparametric data, statistical comparisons were performed using repeated-measures one-way ANOVA with the Kruskal–Wallis test” ─ ANOVA is parametric; Kruskal–Wallis is the nonparametric counterpart. The text conflates them. Needs correction.
Author Response
REVIEWER 2
Comments and Suggestions for Authors
The manuscript explores the immunomodulatory properties of different bacterial lysates in human macrophages, with a particular focus on their ability to restrict M. tuberculosis growth, induce cytokine responses, and stimulate autophagy. The clinical data on bactericidal/permeability-increasing protein (BPI) levels in serum and bronchoalveolar lavage (BAL) fluid of tuberculosis (TB) patients adds value and increases the biomedical relevance. The manuscript is generally well-structured, figures are relevant and well-described. The use of both macrophage models and patient samples strengthens the conclusions. Overall, the manuscript makes a meaningful contribution to the understanding of how bacterial lysates modulate human macrophage responses and how BPI may serve as both a biomarker and an effector molecule in TB. I believe that the manuscript may be published after substantial clarifications and corrections.
Major Comments:
Q1. Lines 249, 251. “335 ±28 µg/mL”, “441 ±49 µg/mL” ─ The unit µg/mL is most likely an error, since the concentration is unusually high. Perhaps pg/mL is intended?.
A1. We thank the reviewer for this important observation. The units were carefully verified, and the correct concentration is nanograms per milliliter (ng/mL). All sections have been revised and corrected accordingly.
Additionally, several studies support IL-8 secretion by monocyte-derived macrophages (MDMs) in the same range as our study’s1, 2
- Varney ML, Olsen KJ, Mosley RL, Bucana CD, Talmadge JE, Singh RK. Monocyte/macrophage recruitment, activation and differentiation modulate interleukin-8 production: a paracrine role of tumor-associated macrophages in tumor angiogenesis. In Vivo. 2002 Nov-Dec;16(6):471-7.
- Victoni, T., Salvator, H., Abrial, C. et al.Human lung and monocyte-derived macrophages differ with regard to the effects of β2-adrenoceptor agonists on cytokine release. Respir Res 18, 126 (2017). https://doi.org/10.1186/s12931-017-0613-y
Q2. Figure 1. The IL-8 concentration on the ordinate axis may contain a unit error (µg/mL). It is likely meant to be pg/mL.
A2. We appreciate the reviewer’s careful reading. The correct unit for IL-8 concentration in Figure 1 is nanograms per milliliter (ng/mL). This has been corrected in both the figure and the text.
Q3. Lines 388, 389, 392. Serum and BAL BPI values. Missing units makes comparison meaningless. It is unclear whether values are pg/mL, ng/mL, or μg/mL. This is a major factual omission.
A3. Thank you for pointing this out. We have now included the correct units in the corresponding sections. The values are expressed as nanograms per milliliter (ng/mL).
Q4. Figure 9. The ordinate axis shows concentrations in µg/mL. Some “half-filled circles” indicate BPI levels of about 2800 µg/mL, which appears unusually high. Please verify the units.
A4: We appreciate the observation. The units were carefully verified, and the correct concentration is nanograms per milliliter (ng/mL). All figures and sections have been revised and corrected.
Minor Comments:
Q5. Line 89. “Isamel Cosio Villegas” ─ Please check, this may be a typographical error.
Line 128. “Pulmonaron” ─ Please check the spelling; earlier and later in the text it appears as “ Pulmonarom”.
Q5. The typographical errors were corrected in the revised version.
Q6. Line 136. “The MNs were seeded” ─ Does this refer specifically to monocytes (MNs) or already to monocyte-derived macrophages (MDMs)? Please clarify.
We appreciate the reviewer’s comment. To avoid confusion, we have replaced the abbreviation MNs with the full term and now consistently use “monocytes” throughout the manuscript.
Q7. Lines 161-162. “anti-human BPI antibody (R&D Systems, Minneapolis, MN, USA)” ─ Please provide more detailed information about the antibody (monoclonal, mouse?).
A7. We added the complete information. The antibody used was a monoclonal mouse IgG2B, anti-human BPI antibody (R&D Systems, Minneapolis, MN, USA)
Q8. Lines 171. “human serum” ─ Please specify the concentration..
A8. We have clarified this point. The concentration used was 10% human serum
Q9. Lines 206. “using 10% BSA” ─ “using 10% BSA solution”.
A9. We thank the reviewer for pointing out this grammatical error. It has been corrected to “using 10% BSA solution”.
Q11. Lines 229-231. “For nonparametric data, statistical comparisons were performed using repeated-measures one-way ANOVA with the Kruskal–Wallis test” ─ ANOVA is parametric; Kruskal–Wallis is the nonparametric counterpart. The text conflates them. Needs correction.
A11. We regret this mistake and have revised the section to correctly describe the statistical methods applied. All results were reanalyzed using the appropriate statistical tests, and the description of the statistical analysis has been corrected accordingly. The applied tests have also been specified in the figure legends for clarity. Importantly, this reanalysis did not alter the study's results or conclusions.

Reviewer 3 Report
Comments and Suggestions for Authors
This manuscript describes a study which evaluated the effects of mycobacterial lysates on human innate immune responses by incubating human peripheral blood monocyte derived macrophages (MDMs) with mycobacterial lysates from commercial sources in comparison with Mycobacterium tuberculosis (Mtb) lab adapted avirulent strain H37Ra by evaluation of cytokine production, intracellular Mtb growth control after infection, autophagy formation and bactericidal/permeability protein (BPI) expression. The results showed that mycobacterial lysates stimulated inflammatory cytokine production by MDMs, and enhanced MDM control of intracellular Mtb growth probably through elevated autophagy mechanisms and production of BPI which was also supported by increased expression of BPI in active tuberculosis patients and its reduction in response to anti-TB therapy. In conclusion, this is an interesting study demonstrating the effects of mycobacterial lysates in enhancing human innate immune responses against Mtb infection with potential mechanism. However, there are some minor concerns which require revision of the manuscript as detailed below:
- It would be better to include brief details of the commercially obtained mycobacterial lysates and their application in previous studies in the introduction.
- I am not sure if MN (monocyte) is a standard abbreviation for monocytes.
- For all the microscopic images shown in Figs 4, 5, 7 and 8, please improve the quality of the images for green channel and need an image in larger magnification so that these markers would be easily visible and to see the differences among different treatments.
- If possible please include the microscopic images of Mtb in MDMs after three day infection for the demonstration of the reduction of Mtb signal in the cells.
- Show the data on the infection rates of Mtb in MDMs after 48 hour mycobacterial lysate stimulation in comparison to the cells without mycobacterial lysate stimulation as separate panel for Fig. 6.
Author Response
REVIEWER 3
Comments and Suggestions for Authors
This manuscript describes a study which evaluated the effects of mycobacterial lysates on human innate immune responses by incubating human peripheral blood monocyte derived macrophages (MDMs) with mycobacterial lysates from commercial sources in comparison with Mycobacterium tuberculosis (Mtb) lab adapted avirulent strain H37Ra by evaluation of cytokine production, intracellular Mtb growth control after infection, autophagy formation and bactericidal/permeability protein (BPI) expression. The results showed that mycobacterial lysates stimulated inflammatory cytokine production by MDMs, and enhanced MDM control of intracellular Mtb growth probably through elevated autophagy mechanisms and production of BPI which was also supported by increased expression of BPI in active tuberculosis patients and its reduction in response to anti-TB therapy. In conclusion, this is an interesting study demonstrating the effects of mycobacterial lysates in enhancing human innate immune responses against Mtb infection with potential mechanism. However, there are some minor concerns which require revision of the manuscript as detailed below:
Q1. It would be better to include brief details of the commercially obtained mycobacterial lysates and their application in previous studies in the introduction.
A1. We recognize the reviewer’s suggestion. In response, we have added information in the Introduction regarding commercially available bacterial lysates and their applications in previous studies. The mycobacterial lysate was prepared in-house, and the method was revised for clarity.
Q2. I am not sure if MN (monocyte) is a standard abbreviation for monocytes.
A2. We appreciate the reviewer’s comment. To avoid any potential confusion, we will use the full term “monocytes” without abbreviations throughout the manuscript. In scientific papers, the term is generally written in full as “monocytes” or “human monocytes,” which we have now adopted to ensure clarity and consistency.
Q3. For all the microscopic images shown in Figs 4, 5, 7 and 8, please improve the quality of the images for green channel and need an image in larger magnification so that these markers would be easily visible and to see the differences among different treatments.
A3. We appreciate the reviewer’s comment regarding the interpretability of the fluorescence images in Figures 4, 5, 7, and 8. In response, we have improved the quality of these figures by increasing their resolution (dpi) to enhance clarity. Moreover, we quantified the percentage of BPI-positive cells (Figures 4 and 6) and LC3-positive cells, a marker of autophagy (Figures 7 and 8). These additions allow us to present the relevant data more clearly.
1) All bacterial lysates significantly increased the expression of BPI
2) Treatment with bacterial lysates increased the proportion of LC3-positive cells, although statistical significance was achieved only with Pulmonarom and Ismigen. Interestingly, when cells pretreated with lysates were subsequently infected with M. tuberculosis, Pulmonarom induced a significant reduction in LC3-positive cells (autophagy), while the other lysates showed no significant effect.
Q4. If possible, please include the microscopic images of Mtb in MDMs after three-day infection for the demonstration of the reduction of Mtb signal in the cells.
A4. We acknowledge the reviewer’s suggestion. Unfortunately, within the limited timeframe provided for the revision, it is not possible to include the requested microscopic images of Mtb in MDMs after three days of infection. However, we will address this point in our ongoing work by performing longitudinal monitoring of the Mtb signal at 0-, 1-, 3-, and 5-day post-infection, which will allow for a more comprehensive assessment of the infection dynamics.
Q5. Show the data on the infection rates of Mtb in MDMs after 48-hour mycobacterial lysate stimulation in comparison to the cells without mycobacterial lysate stimulation as separate panel for Fig. 6.
A5. We thank the reviewer for this thoughtful suggestion. Currently, we do not have data on the infection rates of M. tuberculosis (Mtb) in MDMs after 48-hour stimulation with bacterial lysates, and it is not feasible to generate these results within the current revision period. Nevertheless, we recognize the importance of this analysis, as it could reveal an additional mechanism by which bacterial lysates exert their effects. We will therefore consider incorporating such experiments in our future work.

Round 2
Reviewer 2 Report
Comments and Suggestions for Authors
The authors have substantially revised the manuscript and addressed all of my comments. I believe the manuscript can be accepted for publication in its current form.